# The Impact of Information Flow by Co-Shareholder Relationships on the Stock Returns: A Network Feature Perspective

**DOI:** 10.3390/e24091237

**Published:** 2022-09-02

**Authors:** Pengli An, Sui Guo

**Affiliations:** 1School of Economics and Management, China University of Geosciences, Wuhan 430074, China; 2Center for Energy Environmental Management and Decision-Making, China University of Geosciences, Wuhan 430074, China; 3School of Business, Jiangsu Normal University, Xuzhou 221116, China

**Keywords:** listed energy companies, information flow, stock returns, complex network, structure entropy, panel regression

## Abstract

One shareholder may invest in different listed energy companies, so the information held by common shareholders can be transmitted among companies. Based on the two-mode complex network method, we construct an information flow shareholder-based network and employ different network indicators representing features of information flow as variables to construct panel regression models to analyze the impact of information flow among listed energy companies on the stock returns. The results indicate that the information flow of listed energy companies are increasingly important and play a significant role over a period. The efficiency of information flow among listed energy companies is increasingly high and the network information is concentrated among a few of these companies. The efficiency of information flow and the independence of listed energy companies are significantly positively related to stock returns, while the listed energy companies’ ability to control information is not significantly related to stock returns. We employ a new perspective to analyze the information flow on how to influence stock returns, and offer some related suggestions for investors and policy makers in the future.

## 1. Introduction

Information is very important in the financial markets; past research has used Random Matrix Theory (RMT) and information theory to analyze the correlations and information flow information between news from The New York Times and world financial indices [1] and find the close relationship between them. The information flow between stocks is a universal fact of the global stock market and is well proven in numerous studies [2,3]. Studies have shown that information flows between stocks peak when markets change [4]. In addition, news-driven information spreads between stocks and information flow affects stock market volatility [5,6]. Moreover, the information flows usually influence the investment directions of the entire stock market. Investors gain information and make investment decisions through their professional knowledge and social connections [7,8]. Shareholders are typical investors, in fact, they have access to more information, and have greater sensitivity to information, even disseminate information. Shareholder activity has a significant impact on the information flow around the target company [9], information flows and disseminates among shareholders, especially shareholders who hold the same listed company, namely, common shareholders, usually exchange information [7].

However, previous research on shareholders can focus on various investor behaviors: overreaction [10,11], underreaction [12] and hedging [13,14]. Moreover, relationships among shareholders may have some impacts on companies, some scholars have found shareholder coordination affects stock price [15]. Some scholars have found that when two companies have common shareholders, the competition between the companies is reduced, and when the competition increases, both companies will have better performance [16]. These studies reflect the impact of common shareholders on the company’s performance and stock price from the side; companies can be linked through common shareholders and information can be disseminated based on common shareholders as mentioned above. In this study, we try to study the impact of information flow by co-shareholder relationship on stock returns to rich literature, which is a new perspective to study stock returns.

Energy is an important strategic resource for the country; due to its unique characteristics, listed energy companies are not only an important part of the energy market, but also play a significant role in the energy financial market and energy commodity market [17]. Some scholars have studied the co-holding behavior [17] and the shareholding similarity [18] of the shareholders of listed energy companies. Due to the particularity of listed energy companies, any event in the energy and stock markets ultimately affects shareholders’ decisions, because faster the dissemination of information, the greater impact on shareholders. Shares of company stocks may simultaneously be held by the same shareholders, which connect companies with each other. Information is also transmitted between companies through common shareholders, thus constituting the information network. Many studies have studied the information flow using network method [7,19].

The complex network method is a useful tool that can clearly describe relationships among different subjects and can be applied in various fields, including biology [20,21], medical science [22,23] and economics [24,25]. In the energy financial market, scholars have used complex networks to study the correlations of energy price co-movements [26], the interaction between energy and stock [27], energy companies’ relationships [28], stock market returns [29] and the relationship between shareholders and listed companies [30,31]. In our paper, we use shareholders as medium to study the information flow between listed energy companies and the impact on stock returns. Here, we have two different subjects: listed energy companies and shareholders. Therefore, we use the two-mode network method to research. This method has been applied to study both the stock market [17,32] and social networks [33]. Many scholars also use this method to research the relationships and interactions between listed companies and executives [34,35]. We can construct the co-shareholder relationship network to study the information flow using this method.

We introduce the specific process of the following paper. In this paper, we use data on the ten major circulating shareholders of listed energy companies over twelve years in China to conduct the research, because the decisions of major shareholders have much greater impact on stock returns. Then, we construct co-shareholder relationship networks between listed energy companies with shareholders as the medium based on the heterogeneous two-mode network method, namely, two different subjects, using the decreasing-mode method. In the information flow network embodied by shareholders, we model the listed energy companies as nodes, link each other based on the common shareholders between listed energy companies as edges, and the number of common shareholders between companies as the weights. As the first step in our analysis, we study the evolution of the network to study the spread of information, including the evolution of its nodes, edges and weights; we even study the heterogeneity of nodes in the network through the indicator of network structure entropy. We also study the information flow features in the co-shareholder network, which is reflected by the network indicators: degree, betweenness centrality and closeness centrality of the listed energy companies. We then propose three hypotheses and use panel regression models to analyze whether the information flow among listed energy companies based on shareholders will have an impact on stock returns. Finally, we draw some conclusions based on those results. For convenience, we henceforth use EL-EL network as an abbreviation to represent the co-shareholder behavior network.

Our paper extends the empirical literatures according to studying the impacts of co-shareholder relationships information flow on the stock returns. In doing so, we contribute to the literature in several areas. From the method level, firstly, we construct the co-shareholder relationships between listed energy companies by using common shareholders as a medium to reflect the information flow between companies. Secondly, we use the information flow characteristics reflected by network indicators as independent variables to study the impact on stock returns. From a practical point of view, this research can help investors choose stocks to invest from another perspective, this is novel for investors. In addition, this paper put forward some suggestions to policy makers. For example, they should pay attention to some shareholders not to manipulate the market.

There are four parts in this paper, and the rest of the paper is organized as follows. In Section 2, we introduce the data and methods and propose three hypotheses of the paper. In Section 3, we report the calculation and the analysis of empirical results. The discussion and conclusion are presented in the closing Section 4. 

## 2. Data and Methods

### 2.1. Data

To study the impact of information flow among listed energy companies on stock returns, it is necessary to select some representative stocks. In this paper, we obtain data on the ten major circulating shareholders of listed energy companies and the closing stock price. They are the company’s annual data and daily stock price data, respectively. The data are reported by the Shanghai Stock Exchange and the Shenzhen Stock Exchange and are available from the well-known financial database of China, Wind Data (http://www.wind.com.cn/ (accessed on 17 March 2021)). There are 78 listed energy companies, and there are forty to fifty companies with common shareholder relationships, we used 25 of those in the panel regression, while the market capitalization of them is 83% on all the companies. We use the data on the ten major shareholders to study the information flow feature over a period of twelve years, from 2009 to 2020, in China. In Table 1, we list the code of companies and their market capitalization. This dataset includes the name of the listed energy company, the code of listed energy company and the name of the ten major circulating shareholders as variables. In addition, we need data on stock prices, which consists of several variables: the opening price, closing price, maximum price, and minimum price.

### 2.2. Methods

#### 2.2.1. The Construction of Information Flow Network Embodied by Shareholders

In this study, we construct undirected weighted information flow networks embodied by shareholders. The process of network construction follows previous research [17,36,37] using the decreasing-mode method, which is based on complex network theory. Listed energy companies and shareholders comprise two different sets of actors, and because two given listed energy companies are held by the same shareholder simultaneously, we treat the shareholder as common shareholder; the two companies have co-shareholder relationship, which form an equivalence network [18,38] among listed energy companies. Here, we represent shareholder as an intermediary. In these networks, which we refer to as the EL–EL networks, the nodes represent the listed energy companies, and the edges refer to the information flow represented by co-shareholder relationship between two listed energy companies. In addition, the weight of each edge is represented by the number of common shareholders between listed energy companies.

#### 2.2.2. The Information Flow Represented by Network Indicators

Here, we define and provide mathematical formulas for certain indicators from complex network theory.

(1) Degree

Degree refers to the number of links connecting a node with other nodes in a network, an indicator of the amount of information exchange between a given node and other nodes. Degree can be calculated according to the following formula:(1)$$k_{i}=\sum_{j=1}^{N}a_{ij}$$

Here, $k_{i}$ is the degree of node i, N is the total number of nodes, and $a_{ij}$ refers to the link between nodes i and j.

In the EL–EL network, the degree of a given node is equal to the number of other listed energy companies that have common shareholders with the listed energy company represented by that node. Therefore, degree measures efficiency of information flows that exists between a given listed energy company and other companies in the network, and nodes with higher degrees represent companies with more common shareholders with other companies, the more channels to obtain information, the more efficient the information flow.

(2) Network structure entropy

The network structure entropy represents the difference of network nodes, which is evolved from information entropy. According to previous research, we have learned that the network structure entropy includes many types [39], that is, the nodes, edges and the characteristics of edges, etc. In this paper, we try to study the information flow characteristics of nodes by studying their relative importance. The calculation process is divided into two steps: node relative importance and structure entropy.
(2)$$I_{i}=k_{i}/\sum_{i=1}^{N}k_{i}$$

In the Formula (2), $I_{i}$ indicates the importance of network nodes, $k_{i}$ is the degree of node i, N is the total number of nodes.
(3)$$E=-\sum_{i=1}^{N}I_{i}\ln I_{i}$$

Here, E represents the network structure entropy. The smaller the value of entropy is, the more even the distribution is.

In the EL–EL network, the network structure entropy describes the orderliness of network systems. It is helpful to further research the topology and evolution characteristics of the network by studying the network structure entropy.

(3) Closeness centrality

The measure of closeness centrality describes the extent to which a given node is controlled by other nodes. If the distance between the given node with others is very short, which means that the node is in the center of the network, then the value of the closeness centrality of that node is high. The closeness centrality is calculated using the following formula:(4)$$cc_{i}=\frac{N}{\sum_{j=1}^{N}d_{ij}}$$

Here, $cc_{i}$ is the closeness centrality of node i, N is the total number of nodes, and $d_{ij}$ is the distance from node i to node j. The smaller the value of $d_{ij}$, the larger the value of $cc_{i}$.

In the EL–EL network, the closeness centrality of a node describes how close a listed energy company is to the center of the network. If the closeness centrality of a listed energy company is high, then this company is less reliant on other listed energy companies and is therefore less easily controlled by other listed energy companies; a company’s closeness centrality in the network is positively related to its level of independence, which reflects the independence of listed energy companies from the side, that is to say, the higher the closeness centrality of the given company, the easier it is to obtain information. If there is a company losing contact with the given company, it will not affect his access to information, in other words, it is with higher independence.

(4) Betweenness centrality

The betweenness centrality of a node describes the number of shortest paths that pass through that node and measures the ability of actors to control resources in the network. If a given node is located in the shortest path between many nodes, then the value of betweenness centrality for that node is high, and the actor this node represents will be able to control and distort the transmission of information and influence the construction of the network [38]. The formula in (5) shows the equation for the betweenness centrality of a node:(5)$$b_{\mathrm{ij},e}=\sum_{i}^{N}\;\sum_{j}^{N}\;\frac{f_{\mathrm{ij}}(e)}{f_{\mathrm{ij}}}$$

In the Formula (5), $b_{\mathrm{ij},e}$ is the betweenness centrality of node e, N is the total number of nodes,${\;f}_{\mathrm{ij}}$ is the number of shortest paths between i and j, and $f_{\mathrm{ij}}(e)$ is the number of the shortest paths between i and j through e.

In the EL–EL network, the betweenness centrality of a node measures the mediating role of the listed energy company represented by that node and its ability to control information flow. If any given listed energy company disappears, it will affect the transmission of information in the network depending on its value of betweenness centrality. The greater the value of the betweenness centrality is for a given listed energy company, the greater the ability of that company to control information in the network, what’s more, the disappearance of the company will affect the spread of information throughout the network.

After we construct the EI–EI network, we analyze the information flow among listed energy companies. Then, we build a panel data model and offer three hypotheses to study the impact of information flow among companies on stock returns.

#### 2.2.3. Panel Regression Model and Hypotheses

The data used in the EL–EL network are annual data, so we use data on the annual stock returns of the listed energy company. The stock returns are calculated according to the following formula in (6):(6)$$\mathrm{RA}=\ln P_{\mathrm{last}}-\ln P_{\mathrm{first}}$$

For convenient calculation, we multiplied the stock returns by 100. Next, we construct the panel regression model. The formula for the regression model is shown below:(7)$$y_{t}=b_{0}+b_{1}x_{t1}+b_{2}x_{t2}+\cdots +b_{k}x_{tk}+u_{t}\;\;t=1,2,3,4,5,6,7,8\;\;k=1,2,3,4,5$$

In Formula (7), $y_{t}$ represents the dependent variable, namely, the stock returns of listed energy company, $x_{tk}$ is the independent variable at time *t*, and $b_{k}$ is the coefficient of the independent variable. Here, we have dependent variable, Control variable and Independent variable.

Dependent variables

In the model, stock returns are the dependent variable, which are influenced by general assets and net profits, as well as network indicators representing the information flow of listed energy companies.

Control variables

In the model, the control variables are general assets and net profits. General assets refer to all assets owned or controlled by a listed energy company that can provide economic benefits. The net profits are the main indicator for measuring the business efficiency of a listed energy company.

Independent variables

In the model, the independent variables are the network indicators of listed energy companies, including their degree, closeness centrality, and betweenness centrality. The degree of listed energy companies refers to the efficiency of information flows, which reflects the extent of information flows, the higher the degree value, the more efficient the information flow. The closeness centrality describes the independence of a listed energy company and its ability to remain unaffected by the information flow of other listed energy companies, that is to say, if the value of listed energy company’s closeness centrality is high, information can be received in the given company, and information will not be blocked because one company’s connection with this company is broken. The betweenness centrality of listed energy company refers to the ability of that company to control the information of the network, this indicator is the intermediary of information flow in the EL–EL networks, if the given company disappears, other companies may not link to each other and information flow will be blocked.

In this paper, we propose three hypotheses about the research problem and the panel regression model we have built.

(1) The degree of a listed energy company

Information flows spread between stocks and affect stock market volatility [5,6,40]. Some scholars have studied the relationship between information flow and the prediction of stock returns and have concluded that information flow can predict stock returns [41]. In fact, the amount of information flow and the number of connectedness are different in different periods and events [42]. In addition, information spreads faster in the networks [43]. In our paper, the degree of listed energy companies within the EL–EL network represents the efficiency of information flow between them. The information flows between these companies can indicate information exchange between companies based on shareholders. Consequently, the stock returns of listed energy companies may be affected by the company’s degree in the network.

**Hypothesis** **1.**
*The greater the value of the degree of the listed energy company, the higher the stock returns are.*


(2) The closeness centrality of a listed energy company

Information spillovers have an impact on the stock market, there are many ways to obtain information, so it is in a leading position and will not be affected by a certain channel [44]. Previous studies have also studied the relationship between the independence of the central bank and stock market returns in emerging economies and have concluded that the overall effect of central bank independence on stock market returns is positive [45]. In our paper, the closeness centrality of listed energy companies represents their independence; a given company with a high value of closeness centrality is closely connected with many companies, so a broken connection with a certain company does not affect its access to information, and it has stronger independence. Therefore, we hypothesize that the company’s closeness centrality can impact the stock returns of a listed energy company.

**Hypothesis** **2.**
*The greater the value for the closeness centrality of the listed energy company, the higher the stock returns are.*


(3) The betweenness centrality of a listed energy company

The betweenness centrality of a listed energy company represents that company’s ability to control the information in the network. Some scholars have studied how stock returns are influenced by stock trading through information feedback [46]. There are also researchers studying the source of information, the source of information serves as the information control point, once the information is blocked, the information flows will be cut off [47]. Companies with higher betweenness centrality have stronger control over information, if the company disappears or there is an obstacle to the transmission of information, other companies will not receive the information. In this paper, we study how the ability of listed energy companies to control information affects stock returns. In particular, we hypothesize that the betweenness centrality of a listed energy company positively impacts the stock returns of that company.

**Hypothesis** **3.**
*The greater the value for the betweenness centrality of a listed energy company, the higher its stock returns are.*


Here, in order to clearly demonstrate this part, we use Table 2 to show the variables and hypotheses.

## 3. Results and Discussion

### 3.1. The Evolution of Information Flow Network by Co-Shareholder Relationships

We use data on listed energy companies to construct the information flow network embodied by shareholders according to applying the two-mode network method, a process based on complex network theory. In the networks, listed energy companies comprise the nodes, the common shareholders among companies form the edges, and the number of common shareholders between companies are represented by the weights. Consider the network visibility graphs for 2009 and 2020 as examples, as shown in Figure 1.

In Figure 1, we use the size of the node to represent the degree value of the listed energy company, and the thickness of the line to represent the weight. From Figure 1, we can see that there are more listed energy companies in 2020 than in 2009, indicating that there are more companies with co-shareholder relationships in 2020, and there are more companies with information flow. In addition, there are more nodes with larger degree values in 2020 than in 2009, meaning that a given company has co-shareholder relationships with more other companies. Moreover, there are more edges in 2020 than in 2009, which means more companies with common shareholders, indicating that information flow between companies more frequently in 2020. Furthermore, the thickness of the line in 2020 is thicker than in 2009, indicating that there are more common shareholders between the two companies. In addition to the intuitive graph. We will analyze the evolutionary characteristics of the EL–EL networks by focusing on their nodes, edges and weights according to data, they represent respectively the listed energy companies, the information flow between listed energy companies represented by common shareholders and the number of common shareholders between listed energy companies.

In Figure 2, primary axis shows the changes of nodes, secondary axis shows the evolution of edges and weights. The three curves are rising on the whole over time; however, the fluctuations are very different. In addition, the changes of edges and weights are almost consistent. We explained the changes of the three curves in details. With regard to the nodes, namely listed energy companies, there was an increasing trend from 2009 to 2012 due to the poor conditions in the stock market at that time. China’s resource stocks maintained a high price-to-earnings ratio, and so many listed energy companies received investments by common shareholders. The number of listed energy companies fell in 2013 and 2014 as the result of a bottleneck in the energy market that led to a decline in the profits from resource stocks, and then rose beginning in 2015. From 2016 to 2020, the development of the stock market was stable, and the number of companies did not change much. With regard to the number of edges and weights, namely company pairs with information flows and number of common shareholders, they both reflect the efficiency of information flow, the trends coincide with the trend that the number of listed energy companies displayed, but the number is far higher than that, especially in 2015, mainly because the stock market in China was in a complicated condition in 2015. The stock price soared in the first half of 2015 and reached the highest peak since 2008, which made some major shareholders widely invest, so companies have more common shareholders, and information flow between companies can be through multiple common shareholders. Then, the stock price plummeted in the following months, mainly June and August. The stock price showed an upward trend in July and the last few months of 2015 due to government intervention. The fluctuation trend of stock prices makes shareholders hold a wait-and-see attitude without giving up investing. However, the trends in the number of nodes and edges, weights diverged in 2016 and 2017, when the number of nodes increased while edges and weights declined, mainly because the stock market was not as strong as in the first half of 2015, and major shareholders reduced their investment after information flow. This feature may be explained by an improvement in the stock market, which attracted many retail investors. After all, major shareholders see long-term gains, while retail investors look only at short-term gains. After 2018, the number of edges and weights is on the rise again due to multiple investments by shareholders. It means that the flow of information between companies is more obvious.

In this paper, the average degree represents how many companies have information flow with the given company, the network structure entropy shows the orderliness of information flow between listed energy companies, which further reflects the topological characteristics of the network. Figure 3 shows the evolution characteristics of average degree and network structure entropy from 2009 to 2020.

In Figure 3, the primary axis shows the changes of average degree, and the secondary axis shows the evolution of network structure entropy. On the whole, the evolution trends of the two lines are similar, and both show an upward state, which indicates that the information flow between companies is more and more frequent, and companies in the network are more and more differentiated. Specifically, the network structure entropy had significantly different trend from the average weighted degree in 2012 due to the increase in nodes from 2009 to 2014, indicating that the degree distribution of nodes has strong differences. After 2015, the change trends of network structure entropy and average degree showed a consistent upward trend, indicating that the information flow between nodes in the network was more obvious, and the higher and higher network structure entropy also reflected the uneven distribution and heterogeneity of network nodes.

The results demonstrate that the number of the listed energy companies with information flow and the number of common shareholders representing information flow are increasing. In other words, information flows make stronger and much closer connections between companies, and even the information flow can affect the decisions of common shareholders. The average degree indicates that the information flow between companies is more and more frequent, and the higher and higher network structure entropy indicates that there is a strong difference in the information flow between companies.

### 3.2. The Information Flow Features between Listed Energy Companies in the EL-EL Network

In the EL–EL network, the degree of a given company represents how many other listed energy companies have common shareholder relationship with the company, which reflects the information flows between a given listed energy company and other listed energy companies in the network. The value of degree shows companies have common shareholders and the efficiency of information flow. The closeness centrality of listed energy company indicates its independence, the companies with high value of closeness centrality do not have access to information because of the break with one company. The betweenness centrality of listed energy companies describes its ability to control information; moreover, it measures if the disappearance of the company will affect the spread of information throughout the network. We analyze the characteristics of information flow in the network, and in the twelve-year time span we studied, from 2009 to 2020, only 25 listed companies are present in the data because the common shareholder relationships can change over time. Therefore, we use these 25 listed energy companies in the current study to research the information flow as a basis for later research. As part of this endeavor, we analyze the three network indicators in detail.

We chose 25 listed energy companies from the whole network, which is shown in Figure 4. We used the network in 2015 as an example.

From Figure 4, we can see that the EL–EL network of 25 listed energy companies was randomly selected from the entire EL–EL network. The values of the three network indicators for this sample are also distributed similarly to the entire network, so the sample we selected can effectively represent the characteristics of the whole network.

In Figure 5, we can observe the distribution of degree and closeness centrality in 2009 and 2020 for the two networks, the yellow dots represent the companies in the sample selected from the entire network. The value of the degree for most companies exhibited growth between 2009 and 2020, which indicates the increasing significance of information flow between listed energy companies, and information flows are more efficient. The value of closeness centrality for companies in the network showed a mild decline in 2009 and 2020, which means that the extent of companies’ independence gradually declined and companies have less access to information. According to the figure, it is clear that this degree exhibits a negative relationship with closeness centrality; if a given listed energy company exhibits stronger information flow, that company will also be less independent. As time passes, the disparity in the distribution of degree and closeness centrality is more evident, and the companies in the figure are concentrated at two ends. The results clearly indicate the growth of the degree value for listed energy companies in the network, reflecting the importance of the mutual information between listed energy companies, which play a significant role in the EL–EL network. The value of closeness centrality decreases from 2009 to 2020, and although it is a relative result, this also shows that the independence of the listed energy companies in the EL–EL network is declining, which means that listed energy companies are not able to obtain information because they have no contact with other companies and will not be influenced by other companies.

In Figure 6, we calculate the cumulative distribution of the betweenness centrality of the listed energy companies to show the distributional characteristics and compare the features of the EL–EL network of 25 listed energy companies with the whole network. The yellow dots represent the companies in the sample of 25 companies selected from the entire network. The betweenness centrality of a listed energy company measures its ability to control the information flow of the network. In 2009, the top 20% of the nodes accounted for almost 60% of the betweenness centrality, which means that the top 20% of the listed companies controlled almost 60% of the information. While the situation changed in 2020, at that point, even fewer listed energy companies controlled the information in the network; in particular, the top 20% of the nodes accounted for almost 80% of the betweenness centrality. From Figure 6, we note that the nodes in the cumulative distribution of the EL–EL network of 25 listed energy companies are selected randomly from the entire EL–EL network, especially the turning point. These listed energy companies grasp the information of the entire network established.

These results illustrate that the information in the network is increasingly concentrated among fewer listed energy companies, so the ability of companies in the EL–EL network to control information flows is particularly important.

### 3.3. The Impact of Information Flow by Co-Shareholder Relationship on the Stock Returns

In this paper, we use data from 2009 to 2020 for 25 listed energy companies to construct panel regression models, with stock returns as the dependent variable, general assets and net profits as control variables, and the network indicators of these companies as independent variables, including degree, closeness centrality and betweenness centrality. Before we analyze the results of the regression, we calculate the correlations between all the variables. The results are shown in Table 3.

From Table 3, we can see that the correlation coefficients between most of the variables are very small. There is a significant positive correlation between general assets and net profit, and there is a significant negative correlation between degree and closeness centrality, while closeness centrality and betweenness centrality are significantly positively correlated with stock returns as a whole. The degree, closeness centrality and betweenness centrality indicators are obtained from the EL–EL network. However, the calculation of the three indicators is different, and the meanings represented are also different. Degree of the company means the number of direct links among listed energy companies, which reflects that different shareholders invest in the same company and companies establish links based on shareholders to exchange information. The closeness centrality of companies represents the distance among companies, that is, the company is in the center of the network, and many companies are linked with the company. In the EL–EL network, it means that the given company and many other companies have information flow, if that company in the center. The betweenness centrality of a company means the intermediary of the companies, meaning that if listed energy companies want to contact each other, they need to go through the given company, which reflects that company are in the middle of other companies, and control information flow between other companies. The three indicators are calculated at the different levels, so there is no necessary connection among them. To avoid pseudo regression, we need to test for the stationarity of the data with unit root test before panel estimation. The results of this test show that the variables are stable in second-order differences, so the data are stable. A cointegration test was also conducted, and the results showed that the data exhibit a long-term cointegration relationship.

We used these variables to build three regression models to explain our hypotheses. In the choice of fixed or random effects on the panel regressions, first, we selected the random effects model for estimation, and we performed the Hausman test on the random effects model. The null hypothesis of the Hausman test is that individual effects are independent of regression variables, and random effects models should be established. The *p*-values are all much larger than 0.05 in the three regression models, as showed in Table 4, so we accept the null hypothesis of the Hausman test and choose the random effects in the panel regressions.

The results of the panel regression models we built are shown in Table 5.

Table 5 shows the results of three panel regression models and the influence of each variable on the stock returns. We will analyze these results to explain the impact of information flows of listed energy companies on the stock returns.

Here, we use Table 6 to intuitively display the results of three panel regressions based on three hypotheses.

#### 3.3.1. The Overall Impact of Information Flows between Listed Energy Companies on the Stock Returns

From Table 5 and Table 6, we can see that the information flow of listed energy companies has a significant impact on the stock returns of those companies. In all specifications, the degree of the listed energy companies, namely, the efficiency of information flow, exhibits a significant relationship with stock returns. In models 2 and 3, the closeness centrality of the listed energy companies has a significant impact on stock returns. However, the betweenness centrality of the listed energy companies exhibits no significant relationship with stock returns. It can be seen from the overall results of models 2 and 3 that the degree and closeness centrality of listed energy companies have a positive impact on stock returns, while in the results of model 1, the degree of listed energy companies is negatively correlated with stock returns. In the following section, we will carefully analyze each variable in each model.

#### 3.3.2. The Impact of Each Network Indicator of Listed Energy Companies on the Stock Returns

In model 1 of Table 5 and Table 6, we treat the network indicator of the degree as the independent variable. We then hypothesize that the stock returns of listed energy companies depend on and exhibit a positive relationship with the degree of the listed energy company, which means that the listed energy company with more direct links with other companies have higher stock returns. From the results of model 1, we can see that degree and stock returns are significantly related, but degree does not have a positive impact on stock returns in the EL–EL networks, namely, the smaller the degree value of a listed energy company is, the higher the stock returns of the listed energy company, in other words, information flow between companies negatively affects stock returns, and from a coefficient perspective, the impact extent is small.

In model 2 of Table 5 and Table 6, we treat the network indicators of degree and closeness centrality as the independent variables. The closeness centrality of the listed energy company represents the independence of that company. The hypotheses are that the stock returns of a listed energy company depend positively on the closeness centrality of that company and the that the greater the closeness centrality is, the higher the stock returns are, in other words, if the independence of a listed energy company is strong, it will not be affected by one of other listed energy companies, it would definitely obtain information from a certain company. The results of model 2 show that the degree and closeness centrality of listed energy companies are significantly and positively related to their stock returns. Listed energy companies that maintain more links with other companies and more access to information exhibit higher stock returns. Similarly, listed energy companies that exhibit greater independence also have higher stock returns, according to the coefficient result, which means that listed energy companies in the center of the EL-EL network, that is exchanging information with multiple companies, have higher stock returns.

In model 3 of Table 5 and Table 6, the network indicators of degree, closeness centrality and betweenness centrality are the independent variables, while general assets and net profits are control variables. We hypothesize that the betweenness centrality of a listed energy company will have a positive influence on the stock returns of that company. In other words, stock returns increase with betweenness centrality, so that listed energy companies that control more information flows of the network also have higher stock returns. The result of model 3 shows that both the degree and closeness centrality of listed energy companies are significantly related to their stock returns, but the betweenness centrality of the listed energy company is not. Therefore, while both the degree and closeness centrality of companies positively influence stock returns, no similarly significant relationship exists between stock returns and companies’ ability to control the information flow of the network, since the coefficient is also very small.

#### 3.3.3. The Robustness Test

In order to test the robustness of current results, we select data from 2009 to 2016 to re-do the regression analysis, and we also construct three models, and the results of each model are shown in Table 7.

Table 7 shows the results of three models and the influence of each variable on the stock returns by using different data. By comparing with the results in Table 5, we found that the coefficients of the independent variables and the significance degree of the model have slightly changed, which has no effects on the interpretation of our subsequent results. These results can prove the robustness of the current results.

## 4. Conclusions

There were many studies linking information flows and stock market volatility, mostly starting with the information or news itself. However, information also exists among shareholders. In this paper, we conducted research to analyze the impact of information flow by co-shareholder relationships on stock returns. We constructed EL–EL networks among listed energy companies over twelve years using the decreasing-mode method based on complex network theory. We analyzed the characteristics of information flow according to the network indicators, including the nodes, edges, weights, network structure entropy, degree, betweenness centrality and closeness centrality of listed energy companies, in the EL–EL network to reflect the importance of information flow. Then, we posed three hypotheses and used panel regression models to analyze the impact of information flow on stock returns. We obtained the following results after our detailed analysis:

Firstly, initial results revealed that the number of listed energy companies, as well as company pairs with information flows and number of common shareholders, are increasing, they both reflect the efficiency of information flow, which highlights the importance and increasing significance of information flow among companies. The increasingly close connections between listed energy companies reflect the information flow in EL–EL networks. The companies and the increasing number of common shareholders demonstrate the efficiency of information flow in the networks. Investors and policymakers should pay more attention to the emergence of common shareholders. In particular, policy makers should pay attention to whether the emergence of common shareholders will have the possibility of major shareholders manipulating the market.

Secondly, an important result is that the sample of 25 listed energy companies can represent the characteristics of most listed energy companies in the EL–EL network. We use the evolution of the degree, closeness centrality and betweenness centrality of listed energy companies to explain the importance of information flow. From 2009 to 2020, the degree value of companies in the network increased, reflecting a strengthening common shareholder relationship and efficiency of information flow between listed energy companies. The value of closeness centrality decreased over that period for companies in the network, which means that the listed energy companies in the center of network are less close with other companies and became increasingly less independent over the period, reflecting reduced access to information for companies. The cumulative distribution of betweenness centrality for listed energy companies indicates that the information in the network is controlled by a few companies and that these companies connect the entire network. All the results indicate that the features of information flow of listed energy companies represented by different network indicators are becoming increasingly important and evident. Investors and policy makers should take inspiration from these results. For investors, understanding the characteristics of information flow reflected by network indicators is meaningful for grasping the performance of companies. For policy makers, it is helpful to understand the situation of the company and facilitate real-time supervision.

Moreover, according to the construction of the regression model, we know that there is a direct impact between the stock returns and information flow. In the first model, stock returns and the degree of company are significantly related, while the degree of company does not have a positive impact on stock returns in the EL–EL network. In the second model, a listed energy company’s degree and closeness centrality are significantly positively related to its stock returns, which is the efficiency of information flow and the independence of listed energy companies are significantly related to stock returns. In addition, in the third model, except for the results of the second model, the betweenness centrality of the listed energy companies is not significantly related to its stock returns, which means the information control ability of listed energy companies does not affect their stock returns. These results demonstrate the importance of the closeness centrality of listed energy companies. Therefore, when investors choose stocks for investment, they should pay attention to the closeness centrality of listed energy companies in the information flow network, keeping in mind that the more independent the company is, the higher the stock returns are.

We have provided a new perspective for empirical studies of the energy stock market, and discussed the association between stock returns and information flow of companies based on co-shareholder relationships to reveal the influence factors of stock returns.In addition, we offered a new approach for the study of stock returns using the methods of complex networks and analysis of the characteristics of information flows reflected by different indicators. In future research, we will study how the investment portfolios of listed energy companies impact their stock returns. Meanwhile, we will also consider studying the influence of listed companies’ co-shareholder relationships on stock returns from the perspective of quantitative analysis.

## Figures and Tables

**Figure 1 entropy-24-01237-f001:**
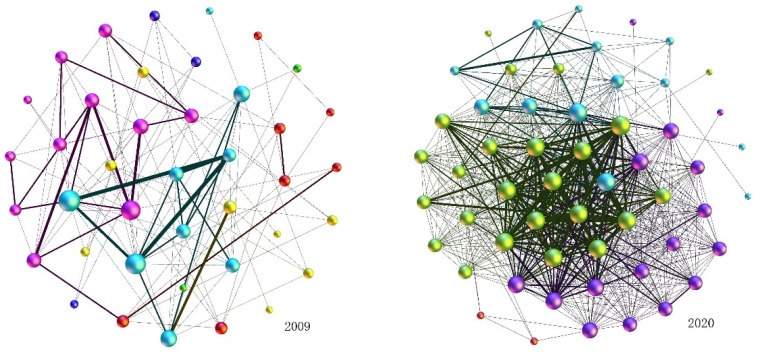
The visibility graph for the information flow network of listed energy companies embodied by shareholders in 2009 and 2020.

**Figure 2 entropy-24-01237-f002:**
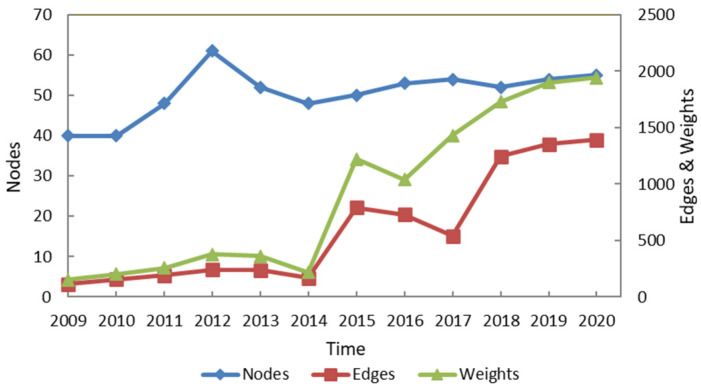
The evolution of nodes, edges and weights from 2009 to 2020 in the EL–EL network.

**Figure 3 entropy-24-01237-f003:**
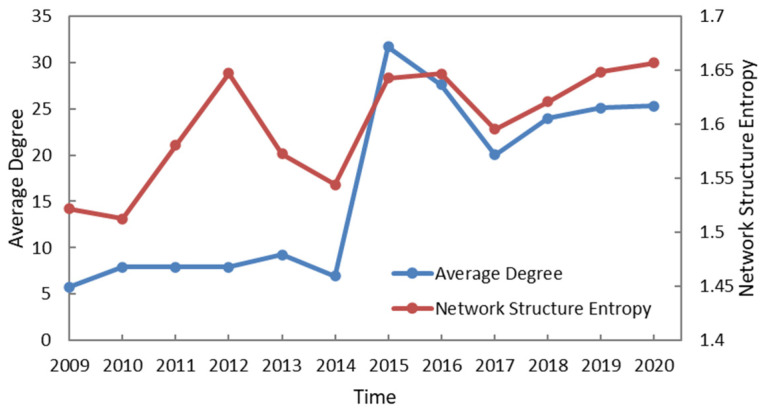
The evolution of network structure entropy from 2009 to 2020 in the EL–EL network.

**Figure 4 entropy-24-01237-f004:**
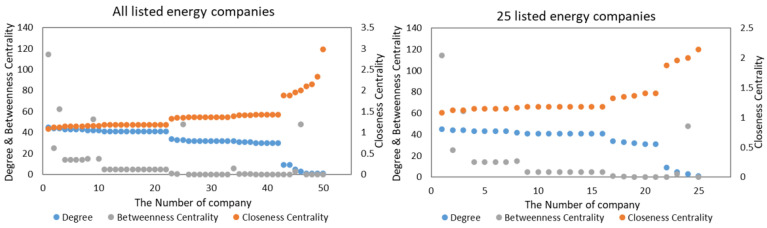
The entire EL–EL network and the EL–EL network of 25 listed energy companies.

**Figure 5 entropy-24-01237-f005:**
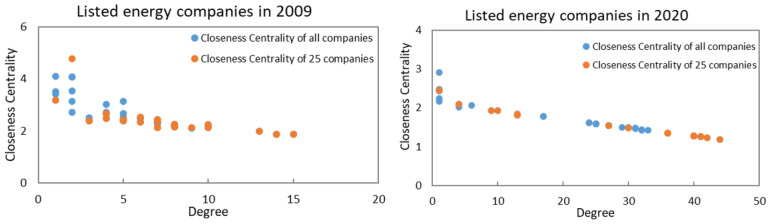
The evolutionary distribution of degree and closeness centrality in entire EL–EL network and the EL–EL network of 25 listed energy companies in 2009 and 2020.

**Figure 6 entropy-24-01237-f006:**
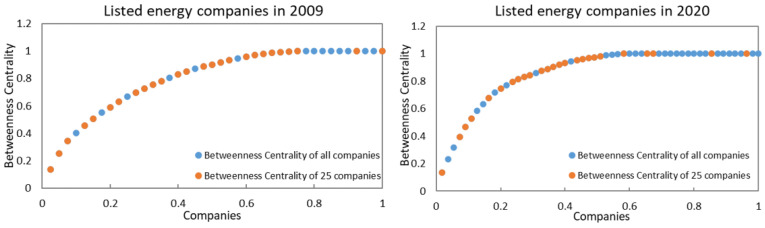
The evolution of the cumulative distribution of betweenness centrality in the entire EL–EL network and the EL–EL network of 25 listed energy companies in 2009 and 2020.

**Table 1 entropy-24-01237-t001:** The codes of representative companies and their market capitalization.

Company	Market Capitalization	Company	Market Capitalization	Company	Market Capitalization	Company	Market Capitalization
000554.SZ	5.36	600188.SH	53.17	600688.SH	69.52	601808.SH	61
000937.SZ	23.94	600256.SH	31.65	600871.SH	78.21	601898.SH	77.08
000968.SZ	10.05	600348.SH	16.21	600971.SH	7.54	60166.SH	11.50
000983.SZ	26.66	600395.SH	13.41	600997.SH	11.42	601857.SH	1453.53
600028.SH	654.09	600397.SH	5.07	601001.SH	10.24	601918.SH	12.19
600123.SH	9.16	600508.SH	7.84	601088.SH	319.79		
600157.SH	49.77	600583.SH	32.78	601699.SH	24.05		

Note: The unit for market capitalization is RMB (billion).

**Table 2 entropy-24-01237-t002:** Summary of hypotheses and multiple variables.

Dependent Variable. Stock Returns	Control Variable		Independent Variable		
	General Assets	Net Profits	Degree	Closeness Centrality	Betweenness Centrality
Hypothesis 1. *efficiency of information flow (degree)*	√	√	√		
Hypothesis 2. *independence (closeness centrality)*	√	√	√	√	
Hypothesis 3. *Information control ability (betweenness centrality)*	√	√	√	√	√

Note: the symbol “√” indicates that the variable is in the model.

**Table 3 entropy-24-01237-t003:** The correlations of variables.

Variables	Mean	S.D.	1	2	3	4	5	6
1. Stock Returns	−5.795052	44.43605	1					
2. General Assets	20,353,714	50,879,771	−0.0042	1				
3. Net Profits	844,420.1	2,256,781	0.0244	0.8312	1			
4. Degree	20.24000	14.14151	−0.1278	0.1760	0.0736	1		
5. Closeness Centrality	1.815812	0.501725	0.2247	−0.1624	−0.0825	−0.8884	1	
6. Betweenness Centrality	29.98618	38.00608	0.0693	0.0763	0.1189	0.0394	−0.0481	1

**Table 4 entropy-24-01237-t004:** Hausman specification test.

Model	Chi-Sq. Statistic	Chi-Sq. d.f.	Prob.
Model 1	1.507650	3	0.6805
Model 2	6.666736	4	0.1546
Model 3	5.672577	5	0.3394

**Table 5 entropy-24-01237-t005:** Results of the panel regression.

Variables	Model 1	Model 2	Model 3
General Assets	−2.81 × 10^−8^ (−0.303675)	−4.06 × 10^−8^ (−0.451809)	−3.48 × 10^−8^ (−0.387413)
Net Profits	1.19 × 10^−6^ (0.578055)	1.60 × 10^−6^ (0.800870)	1.32 × 10^−6^ (0.657425)
Degree	−0.397639 ** (−2.143442)	1.093366 *** (2.833822)	1.094130 *** (2.839841)
Closeness Centrality		47.20543 *** (4.369654)	47.54138 *** (4.405863)
Betweenness Centrality			0.089400 (1.356995)

Note: Unstandardized coefficients are reported with t values in parentheses. ** *p* < 0.05; *** *p* < 0.01.

**Table 6 entropy-24-01237-t006:** Summary of Hypotheses and multiple variables.

Dependent Variable. Stock Returns	Control Variable		Independent Variable		
	General Assets	Net Profits	Degree	Closeness Centrality	Betweenness Centrality
Hypothesis 1. *efficiency of information flow (degree)*	+	−	− **		
Hypothesis 2. *independence (closeness centrality)*	−	+	+ ***	+ ***	
Hypothesis 3. *Information control ability (betweenness centrality)*	−	+	+ ***	+ ***	+

Note: the symbol “+” and “−” indicates the positive and negative impact of the given variable to stock returns. ** *p* < 0.05; *** *p* < 0.01.

**Table 7 entropy-24-01237-t007:** Results of the panel regression for different length of time.

Variables	Model 1	Model 2	Model 3
General Assets	5.81 × 10^−8^ (0.351548)	−3.22 × 10^−8^ (−0.202623)	−2.31 × 10^−8^ (−0.144665)
Net Profits	−6.89 × 10^−7^ (−0.217639)	1.26 × 10^−6^ (0.412163)	8.68 × 10^−7^ (0.282561)
Degree	−0.742132 ** (−2.324114)	1.591635 ** (2.344781)	1.615970 ** (2.375856)
Closeness Centrality		66.97370 *** (3.844647)	68.16986 *** (3.901876)
Betweenness Centrality			0.119855 (1.362799)

Note: Unstandardized coefficients are reported with t values in parentheses. ** *p* < 0.05; *** *p* < 0.01.

## Data Availability

Not applicable.

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
