# Peer review of "The Impact of Information Flow by Co-Shareholder Relationships on the Stock Returns: A Network Feature Perspective"

_entropy, 2022, doi:10.3390/e24091237_

Round 1

Reviewer 1 Report

This paper studies the influence of information flow by co-shareholder relationships on the stock returns of Chinese exchanges. Overall, the paper is well written. I only have a few minor comments.

1) Could the authors please specify what is the frequency of data the used? Is it intraday or daily data?

2) In relation to last point, I am wondering if the robustness of current results could be further checked from a methodological side and an empirical side. For the later, for instance, would daily or intraday data significantly change their findings?

3) Figure 1 is not quite readable. I’m not sure if there is a better way, but the current choice is not quite acceptable.

Reviewer 2 Report

In this paper, authors build a complex network based on information flow from shareholders, to use on a panel regression framework to analyse information flow between listed energy companies and stock returns. I found the paper interesting, although I believe it could be improved paying attention to the following questions:

1. I believe that authors could deepen their literature review, which in their case is included on the introduction, to reinforce the objectives and added value of their paper.

2. The method is well described, although authors should care with the presentation of subtitles like "Dependent variable" (line 225), "Control variable" (line 229) and so on.

3. In the conclusions, authors should reflect on the implications for the different agents, which are not clear in the text. This will be important for the paper.

4. Also in the last section, consider the hypothesis of including, in future research the possible implication of multifractality in the analysis (see, for example, https://www.sciencedirect.com/science/article/pii/S0378437117305642).

5. Be sure that all referenced papers are in the final list, as well as the contrary.
